# Fully Automated Measurement of GFAP in CSF Using the LUMIPULSE^®^ System: Implications for Alzheimer’s Disease Diagnosis and Staging

**DOI:** 10.3390/ijms26178134

**Published:** 2025-08-22

**Authors:** Hisashi Nojima, Mai Yamamoto, Jo Kamada, Tomohiro Hamanaka, Katsumi Aoyagi

**Affiliations:** FUJIREBIO INC., 1-8-1 Akasaka, Minato-ku, Tokyo 107-0052, Japan; mai.yamamoto@hugp.com (M.Y.); jo.kamada@hugp.com (J.K.); tomohiro.hamanaka@hugp.com (T.H.); katsumi.aoyagi@hugp.com (K.A.)

**Keywords:** GFAP, Alzheimer’s disease, CLEIA

## Abstract

Glial fibrillary acidic protein (GFAP) has been shown to be a reliable biomarker for detecting neurological disorders. Recently, we developed the Lumipulse *G* GFAP plasma assay, which is a commercially available tool. Compared to existing assays, the LUMIPLSE *G* platform offers the high-throughput, rapid, and fully automated quantification of biomarkers, enabling more standardized and accessible clinical study. In this study, we evaluated this assay using cerebrospinal fluid (CSF) samples. Assessing GFAP in CSF may provide more direct insights into central nervous system pathology than plasma and could improve the characterization of Alzheimer’s disease (AD) stages and support treatment monitoring. The LUMIPULSE G system is a chemiluminescent enzyme immunoassay (CLEIA) platform equipped with full automation, utilizing specialized cartridges to process samples within 30 min. The assay, which employs a pair of proprietary monoclonal antibodies targeting GFAP, was evaluated for clinical performance using 30 CSF samples from patients diagnosed with AD, patients with mild cognitive impairment (MCI), and cognitively unimpaired (CU) individuals, with 10 samples from each group. In addition, levels of β-amyloid 1–40 (Aβ40), β-amyloid 1–42 (Aβ42), and pTau181 were simultaneously measured. The Lumipulse *G* GFAP assay significantly differentiated (*p* < 0.05) between the amyloid accumulation and non-amyloid accumulation groups, as classified based on the CSF Aβ test. Furthermore, GFAP showed a moderate correlation with pTau181 (r = 0.588), as determined based on Spearman’s rank correlation coefficient. Moreover, receiver operating characteristic (ROC) analysis was performed to determine the performance of GFAP in distinguishing amyloid-positive and amyloid-negative subjects, with an area under the curve (AUC) of 0.72 (0.50–0.93). When stratified by CSF pTau181 positivity, GFAP demonstrated an improved diagnostic accuracy, achieving an AUC of 0.86 (95% CI: 0.68–1.00). This study demonstrates that the Lumipulse *G* GFAP assay, when applied to CSF samples, has the potential to differentiate AD from non-AD cases, particularly suggesting its utility in detecting tau-related pathology. While GFAP has previously been established as a biomarker for AD, our findings highlight that combining GFAP with other biomarkers such as Aβ40, Aβ42, and pTau181 may enhance the understanding of AD pathogenesis, disease staging, and possibly treatment responses. These findings suggest that GFAP may serve as a complementary biomarker reflecting astroglial reactivity associated with tau positivity, alongside established biomarkers such as Aβ40, Aβ42, and pTau181. However, since GFAP levels may also be elevated in other neurological disorders beyond AD, further investigation into these conditions is required.

## 1. Introduction

Alzheimer’s disease (AD), the most prevalent form of dementia, is characterized by the extracellular deposition of β-amyloid (Aβ) plaques and the intracellular accumulation of neurofibrillary tangles composed of hyperphosphorylated tau [1]. These pathological hallmarks contribute to progressive neuronal loss, synaptic dysfunction, and, ultimately, cognitive decline in affected individuals. It is estimated that approximately 6.9 million Americans aged 65 and older are living with AD in 2024, and projections suggest that this number will rise to around 13.8 million by 2060 [2].

In addition to amyloid and tau pathology, AD is also characterized by neuroinflammation and the activation of astroglial cells [3,4]. Astrocytes, a major type of glial cell, respond to central nervous system (CNS) injury through astrogliosis, a process marked by the overexpression of glial fibrillary acidic protein (GFAP), a type III intermediate filament protein specific to mature astrocytes [5]. Under pathological conditions such as AD, reactive astrocytes surrounding Aβ plaques significantly upregulate GFAP expression, which has emerged as a key biomarker for monitoring astrocytic responses [6].

Recent evidence suggests that GFAP levels may also reflect therapeutic responses to anti-amyloid treatments. Indeed, a significant decrease in plasma GFAP levels from baseline was observed after 12 weeks of donanemab treatment compared with the placebo, and this reduction persisted throughout the 76-week study period [7]. These observations suggest that the reduction in the amyloid burden induced by anti-amyloid antibody therapy is associated with a concomitant decrease in GFAP levels, highlighting its potential utility as a treatment-responsive biomarker.

Historically, CSF has served as the gold standard biological fluid for detecting neuropathologic biomarkers due to its proximity to brain tissue, with AD biomarkers such as decreased Aβ42 and increased total Tau (t-Tau) and phosphorylated Tau (p-Tau) [8,9]. GFAP has been studied as a marker of neuroinflammation and reactive astrogliosis [3,4]. Notably, some studies have found that plasma GFAP shows stronger associations with the Aβ burden, cognitive decline, and conversion from mild cognitive impairment (MCI) to AD than CSF GFAP [10,11]. A meta-analysis of 34 studies reported pooled diagnostic accuracy for blood GFAP in distinguishing AD from controls, which was higher than that of CSF GFAP, although its performance improved when combined with demographic factors such as sex, age, and the *APOE* genotype [12]. However, GFAP levels in CSF may provide a more direct representation of central nervous system pathology, especially in the early disease stages, as they are less influenced by peripheral factors affecting blood-based biomarkers. Thus, CSF GFAP may complement plasma measurements by offering greater specificity for brain-derived astrocytic activity.

While GFAP assays have demonstrated utility in research, their translation into routine clinical use is currently limited by factors such as assay complexity, cost, and throughput. Recently, we developed a fully automated blood measurement kit for GFAP using the LUMIPULSE platform [13]. This kit is designed to improve the accessibility and efficiency of GFAP testing in research settings, offering rapid and reproducible results suitable for exploratory and preclinical studies. By automating the entire testing process, we aim to increase throughput, reduce human error, and ensure consistent results. Our GFAP measurement kit utilizes highly specific antibodies and a streamlined protocol to deliver accurate and reproducible measurements [13]. Furthermore, the LUMIPULSE platform offers a comprehensive portfolio of AD-related biomarkers, facilitating sequential measurements and comprehensive analyses of neurological conditions [8,13,14,15].

In this study, we aimed to evaluate CSF GFAP levels across the AD spectrum, including cognitively unimpaired individuals, patients with MCI, and those with clinically manifest AD. We further investigated the relationship between GFAP concentrations measured using our novel assay and established CSF biomarkers, in order to determine its diagnostic and pathophysiological relevance.

## 2. Results

### 2.1. Specimens and Biomarker Characteristics

Table 1 summarizes the demographic characteristics and biomarker results of the study specimens. A total of 30 specimens were used (10 CU, 10 MCI, and 10 AD). There were no significant differences in age or sex among the clinical groups.

### 2.2. GFAP Levels in Relation to Clinical Diagnosis and Core AD Biomarkers

Based on the CSF Aβ42/Aβ40 ratio, 0 CU (0%), 2 MCI (20%), and 9 AD (90%) samples were classified as amyloid-positive. The AD group showed a significant difference compared to both the CU and MCI groups, whereas there was no statistically significant difference between the CU and MCI groups (Figure 1).

Then, we analyzed GFAP levels across the clinical stage groups and found no significant differences among them (Figure 2).

However, when we compared GFAP levels between CSF Aβ42/Aβ40-ratio-positive and negative groups, as well as between CSF pTau181-positive and negative groups, we found that GFAP levels were significantly different based on Aβ and pTau positivity (Figure 3). The cutoff values used to define biomarker positivity were 0.067 for the Aβ42/Aβ40 ratio, based on a previously published study [8], and 56.5 pg/mL for pTau181, as specified in the manufacturer’s package insert. Based on these thresholds, 11 specimens (36.7%) were classified as Aβ-positive, and 10 specimens (33.3%) were classified as pTau181-positive.

We further analyzed GFAP levels in relation to the CSF Aβ42/Aβ40 ratio and CSF pTau181 and found that GFAP levels showed a weak negative correlation with the CSF Aβ42/Aβ40 ratio (ρ = –0.368, *p* = 0.045) and a moderate positive correlation with CSF pTau181 (ρ = 0.588, *p* = 6.358 × 10^−4^) (Figure 4).

Partial correlation analyses demonstrated no significant association between GFAP and the Aβ42/Aβ40 ratio after adjusting for age alone (ρ = –0.230, *p* = 0.230) or for both age and sex (ρ = –0.233, *p* = 0.232). In contrast, GFAP was significantly positively correlated with pTau181 after adjusting for age (ρ = 0.564, *p* = 0.001) and also when adjusting for both age and sex (ρ = 0.551, *p* = 0.002) (Table 2). 

### 2.3. CSF GFAP Discriminates Aβ and pTau Positivity: ROC Curve Analysis

To evaluate the discriminative performance of CSF GFAP, we conducted receiver operating characteristic (ROC) curve analyses using dichotomized groups based on the CSF Aβ42/Aβ40 ratio and CSF pTau181 status. As shown in Figure 5, GFAP demonstrated a moderate ability to distinguish Aβ42/Aβ40 ratio-positive cases from negative ones, with an area under the ROC curve (AUC) of 0.72 (95% CI: 0.50–0.93) (Figure 5A). On the other hand, GFAP showed even higher discriminative performance for CSF pTau181 positivity, yielding an AUC of 0.86 (95% CI: 0.68–1.00) (Figure 5B). Cutoff values were determined using Youden’s index (Table 3). Sensitivity and specificity are presented with 95% confidence intervals (95% CIs) shown in parentheses, calculated using the exact method (Table 3).

## 3. Discussion

This study demonstrates the clinical utility of a fully automated GFAP assay in CSF samples using the LUMIPULSE platform. Although CSF GFAP levels did not significantly differ across clinical diagnostic groups (CU, MCI, and AD), they showed robust associations with core AD biomarkers. The monoclonal antibodies used in the assay target the core region of GFAP, ensuring high specificity and minimizing cross-reactivity with other intermediate filaments such as neurofilaments [13]. Therefore, the observed signal is unlikely to be attributable to cross-reactivity with structurally similar proteins. One possible explanation for the lack of significant group differences is that conventional clinical classifications may not adequately reflect the underlying biological pathology of AD [4,16]. These categories are primarily based on cognitive symptoms, which can be influenced by various non-pathological factors such as comorbidities, and do not always align precisely with the stage or presence of AD-related pathology. This underscores the importance of incorporating biomarker-based frameworks, such as the A/T/N classification, which allow for a more accurate staging of the disease continuum by capturing amyloid deposition (A), tau pathology (T), and neurodegeneration (N), independent of clinical presentation [17]. Such biologically grounded approaches may provide a more reliable context in which to interpret fluid biomarkers like GFAP.

Several previous studies have reported elevated CSF GFAP levels in AD patients, often associated with Aβ and tau pathology [18,19]. However, some studies have demonstrated that CSF GFAP may be unstable after multiple freeze–thaw cycles [20]. Additionally, Cicognola et al. reported that plasma GFAP could predict progression from MCI due to AD and detect abnormal CSF Aβ and tau status [11]. These findings align with our observations, reinforcing the role of GFAP as a biomarker reflecting astroglial activation associated with AD pathology.

The moderate positive correlation between CSF GFAP and pTau181 (ρ = 0.588) suggests that GFAP primarily reflects tauopathy-associated astrogliosis rather than amyloid pathology. After adjusting for age and sex, the significant association between CSF GFAP and pTau181 remained, suggesting that the relationship is not confounded by demographic factors. This supports the robustness of our findings. This finding aligns with histopathological evidence that reactive astrocytes preferentially cluster around neurofibrillary tangles and degenerating neurons, releasing pro-inflammatory cytokines and promoting further tau phosphorylation and neurodegeneration [3,6,21].

The superior discriminative performance of GFAP for tau pathology (AUC = 0.86) over amyloid pathology (AUC = 0.72) implies that astrocyte reactivity is more closely linked to tau-driven neuronal injury. This may help explain why GFAP elevations emerge later in the AD continuum compared to amyloid biomarkers, coinciding with the onset of overt neurodegeneration.

While several studies report that plasma GFAP correlates more strongly with the amyloid burden compared to CSF GFAP [10,12], our data emphasize the potential of CSF GFAP to reflect tauopathy-related processes. The weaker amyloid association in CSF may be due to the fact that plasma GFAP integrates systemic inflammation and blood–brain barrier (BBB) integrity, whereas CSF GFAP more directly reflects central astrocyte activity [10]. This concept is consistent with the revised diagnostic framework for AD, which proposes that amyloid pathology precedes tau-mediated neurodegeneration [4]. In this context, astrocyte reactivity may initially be triggered by amyloid deposition, but persistent GFAP elevation likely requires ongoing tau pathology and neuroinflammation [6].

This study has several limitations that warrant consideration. First, the modest sample size (*n* = 30) limits the statistical power and restricts the ability to perform robust subgroup analyses across the AD continuum (CU/MCI/AD), thereby hindering the detection of stage-specific biomarker differences. Additionally, the small sample size may affect the stability and precision of diagnostic performance estimates, including the sensitivity, specificity, and AUC in ROC analyses. Indeed, post hoc power analyses revealed limited power, estimated at approximately 38% for the comparison between Aβ-positive (*n* = 11) and Aβ-negative (*n* = 19) groups and approximately 37% for the comparison between pTau181-positive (*n* = 10) and pTau181-negative (*n* = 20) groups. These results reflect the exploratory nature of the study and emphasize the need for larger, adequately powered studies. Moreover, although we applied 1000-fold bootstrap resampling to enhance the robustness of the AUC CIs, the potential for statistical variability remains. Future studies involving larger, longitudinal cohorts are needed to validate these findings and examine the temporal dynamics of GFAP in relation to other biomarkers. Second, while CSF GFAP was significantly associated with tau pathology, its disease specificity remains limited. GFAP is elevated in a variety of neurological disorders including traumatic brain injury (TBI), stroke, and neuroinflammatory conditions [8,22,23,24,25], potentially confounding its diagnostic utility in AD. Subsequent studies should directly compare AD with non-AD dementias to clarify the discriminative capacity of CSF GFAP. Most critically, our findings underscore unresolved questions regarding the functional heterogeneity of astrocytes in AD. While GFAP is a widely used marker of astrocyte reactivity, it does not distinguish between the diverse phenotypes of reactive astrocytes, including potentially neuroprotective versus neurotoxic states [6]. Future research should integrate single-cell transcriptomic profiling with longitudinal GFAP measurements to better understand the role of astrocytes in AD. Specifically, it will be important to investigate how GFAP trajectories correlate with distinct astrocyte subpopulations, whether threshold levels of GFAP can be established to differentiate between compensatory and detrimental astrogliosis, and how disease-modifying interventions influence astrocyte phenotype transitions over time. Lastly, the absence of CSF pTau217 and Neurofilament light chain (NFL) measurements represents another limitation. Although pTau181 was included as a representative tau biomarker, recent studies suggest that pTau217 may offer superior diagnostic performance in AD. Similarly, CSF NFL is considered a key marker of neurodegeneration (“N” in the A/T/N framework). However, due to insufficient sample volume, we were unable to include pTau217 or NFL in this study. Future investigations incorporating these markers with newly collected samples will be important for further validation and comparison.

## 4. Materials and Methods

### 4.1. CSF Samples

CSF samples were obtained from the NCNP Biobank (National Center of Neurology and Psychiatry, Tokyo, Japan). A total of 30 samples were analyzed, including 10 from CU individuals, 10 from patients with MCI, and 10 from patients with AD. All participants provided written informed consent, and the study was approved by the institutional review boards of the NCNP and H.U. Group Holdings, Inc., under approval numbers NCNPBB-0091 and 19-011-06, respectively.

### 4.2. Sample Analysis

CSF concentrations of GFAP, Aβ40, Aβ42, and pTau181 were measured using the fully automated LUMIPULSE G 1200 analyzer (FUJIREBIO INC., Tokyo, Japan) as previously described [8,13]. Briefly, the assays are based on a two-step CLEIA principle. In the first step, target-antibody-coated magnetic capture beads are combined with the sample. Target molecules present in the sample are captured by the antibody-coated capture beads. After washing, alkaline-phosphatase-labeled antibodies are mixed with the capture beads. The detector antibodies bind to the captured target. Following a second wash, the capture beads are resuspended, and 3-(2′-spiroadamantyl)-4-methoxy-4-(3″-phosphoryloxy)-phenyl-1,2-dioxetane (AMPPD) substrate is added to the reaction mixture. The resulting luminescence is measured at 477 nm. The intensity of the luminescent reaction is directly proportional to the concentration of the target in the test sample. Results are available within approximately 30 min.

### 4.3. GFAP Assay Discription

For GFAP measurements, CSF samples were diluted 10-fold using the Lumipulse Specimen diluent 1 prior to analysis. The lower limit of quantification (LLOQ) in the diluted CSF matrix was 5.9 pg/mL. Spike-and-recovery experiments showed recovery rates ranging from 94% to 98%, and dilution linearity was confirmed up to 16-fold dilution, with recoveries between 94% and 103%. No high-dose hook effect was observed up to the assay’s upper measurement limit of 5000.0 pg/mL, or even at concentrations up to 7231.7 pg/mL. The stability of diluted CSF samples was maintained for at least up to 2 freeze–thaw cycles. Within-run and between-run precision (coefficient of variation, CV%) were ≤2.5%, and ≤1.7%, respectively.

### 4.4. Statistics

All statistical analyses were performed with R version 4.4.3 (R Foundation for Statistical Computing, Vienna, Austria). Continuous variables were summarized as the median, first quartile (Q1), and third quartile (Q3). Group comparisons for continuous variables were performed using the Wilcoxon rank-sum test and the Kruskal–Wallis test, with Bonferroni correction applied where appropriate. Categorical variables were compared using Fisher’s exact test. Spearman correlation coefficients (ρ) were calculated. Partial correlation analyses were conducted to examine the associations between GFAP and the Aβ42/Aβ40 ratio, as well as between GFAP and pTau181, while adjusting for age and sex as covariates. Receiver operating characteristic (ROC) analyses were performed to evaluate the performance of individual biomarkers in discriminating Aβ positivity and/or Tau positivity. Post hoc power analyses were conducted using G*Power [26] (ver. 3.1.9.7; Heinrich-Heine-Universität Düsseldorf, Düsseldorf, Germany) to assess the statistical power of the current sample, assuming a two-tailed *t*-test, α = 0.05, and an effect size of Cohen’s d = 0.65. The significance level for the statistical analyses was set at *p* < 0.05.

## 5. Conclusions

This fully automated GFAP assay using CSF samples reliably detects tau-associated astrogliosis in AD. The LUMIPULSE G 1200 analyzer enables rapid (30 min) and high-throughput (up to 120 tests per hour) analysis [27]. This level of automation reduces pre-analytical variability and allows for the simultaneous quantification of core AD biomarkers (Aβ40, Aβ42, and pTau181), thereby supporting comprehensive and efficient pathological profiling [8,13]. Although CSF collection remains an invasive procedure [28], integrating GFAP with other established biomarkers on the LUMIPULSE platform provides a standardized and scalable tool for monitoring neurodegenerative changes in therapeutic trials and for aiding differential diagnosis in clinical settings. Moreover, these findings support the view that GFAP reflects astroglial activation in AD and may complement established core biomarkers. GFAP is increasingly recognized as a reliable biomarker for both the diagnosis and staging of AD, with its measurement in CSF or plasma offering valuable insight into disease pathology and progression.

## Figures and Tables

**Figure 1 ijms-26-08134-f001:**
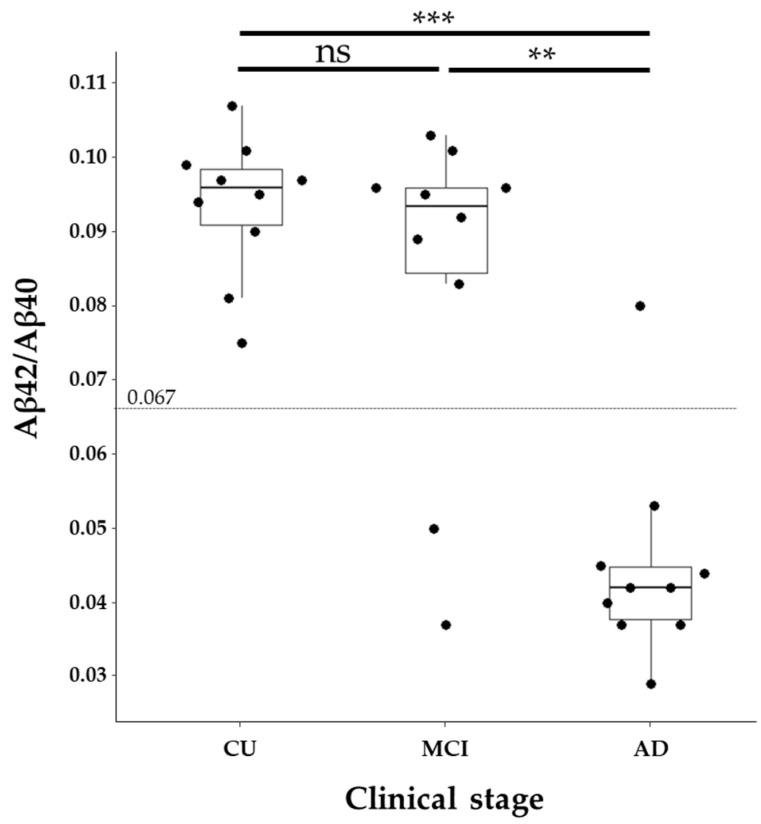
Distribution of Aβ42/Aβ40 ratios across different groups. The figure presents a beeswarm boxplot of Aβ42/Aβ40 ratios. The boxplot displays the median along with the interquartile range (IQR), represented by the lower and upper hinges, and whiskers extending to ±1.5 times the IQR from the first and third quartiles. Data were analyzed using the Wilcoxon rank-sum test with a Bonferroni correction. ns: not significant; **: *p* < 0.01; ***: *p* < 0.001.

**Figure 2 ijms-26-08134-f002:**
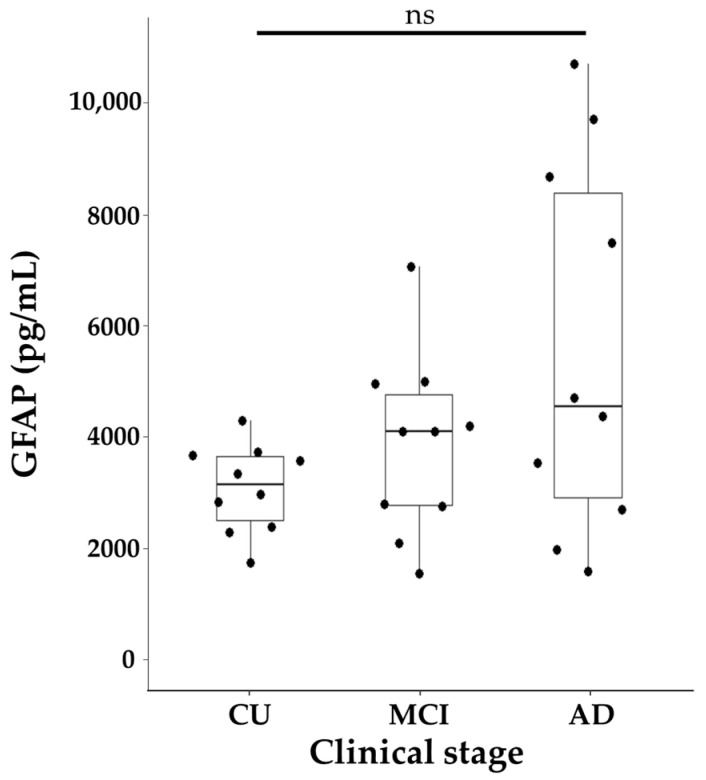
Distribution of GFAP across different groups. The figure presents a composite plot including beeswarm plots and boxplots of GFAP. The boxplot displays the median along with the interquartile range (IQR), represented by the lower and upper hinges, and whiskers extending to ±1.5 times the IQR from the first and third quartiles. Data were analyzed using the Wilcoxon rank-sum test with a Bonferroni correction. ns: not significant.

**Figure 3 ijms-26-08134-f003:**
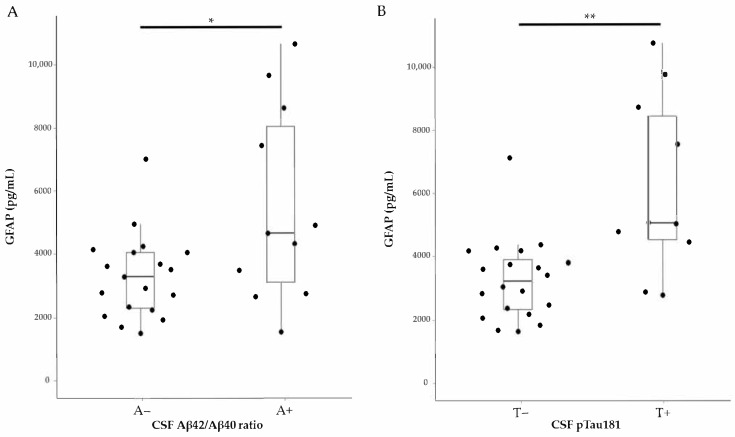
Distribution of GFAP by biomarker-positive and negative groups. The figure presents a composite plot including beeswarm plots and boxplots of GFAP. (**A**) Distribution based on CSF Aβ42/Aβ40 ratio positivity. (**B**) Distribution based on CSF pTau181 positivity. The boxplot displays the median along with the interquartile range (IQR), represented by the lower and upper hinges, and whiskers extending to ±1.5 times the IQR from the first and third quartiles. Data were analyzed using the Wilcoxon rank-sum test. ns: not significant; *: *p* < 0.05; **: *p* < 0.01.

**Figure 4 ijms-26-08134-f004:**
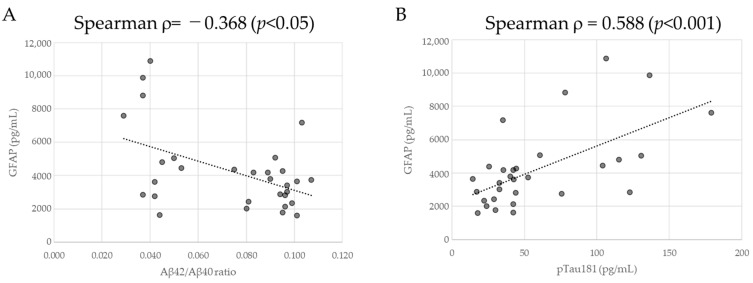
Correlation between GFAP concentration and (**A**) CSF Aβ42/Aβ40 ratio or (**B**) CSF pTau181. Black dashed linear regression lines are superimposed on scatter plots. Black dots represent individual samples. Spearman’s rank correlation coefficient was used for statistical analysis, and *p*-values are indicated.

**Figure 5 ijms-26-08134-f005:**
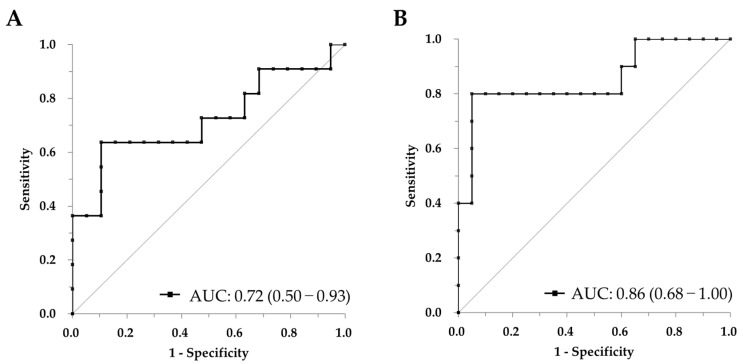
Receiver operating characteristic (ROC) curves for CSF GFAP in relation to (**A**) CSF Aβ42/Aβ40 ratio and (**B**) CSF pTau181 levels. For the ROC analysis, samples were dichotomized into biomarker-negative and biomarker-positive groups based on each CSF marker. The area under the ROC curve (AUC) was calculated to assess the performance of GFAP in distinguishing between these groups. Values in parentheses indicate the 95% CIs. The black diagonal line represents the reference line, which indicates no discrimination (i.e., an AUC of 0.5).

**Table 1 ijms-26-08134-t001:** Characteristics of all specimens by clinical status.

	**CU**	**MCI**	**AD**	***p*-Value**
*n*	10	10	10	
Gender, Female, *n* (%)	5 (50.0)	5 (50.0)	6 (60.0)	1
Age, years *	63.50 [61.25, 66.75]	70.50 [63.50, 80.25]	72.50 [68.50, 77.25]	0.182
Aβ40 *	9262 [7604, 11,847] pg/mL	10,668 [10,158, 12,705] pg/mL	9103.5 [6775, 11,239] pg/mL	0.272
Aβ42 *	814 [620, 1136] pg/mL	981 [670, 1003] pg/mL	392 [319, 418] pg/mL	<0.001
Aβ42/Aβ40 ratio *	0.096 [0.091, 0.098]	0.094 [0.084, 0.096]	0.042 [0.038, 0.045]	<0.001
pTau181 *	29.6 [23.0, 32.7] pg/mL	43.1 [37.2, 56.6] pg/mL	91.0 [51.0, 113.0] pg/mL	<0.05
GFAP *	3212.5 [2549.0, 3710.5] pg/mL	4176.5 [2825.0, 4847.2] pg/mL	4621.5 [2970.2, 8523.2] pg/mL	0.244

* Median [Q1, Q3]. *p*-values: Kruskal–Wallis test for continuous variables; Fisher’s exact test for sex.

**Table 2 ijms-26-08134-t002:** Spearman partial correlation coefficients between GFAP and the Aβ42/Aβ40 ratio or pTau181, adjusted for age and sex.

Variable A	Variable B	Adjusted Variables	r (Partial)	*p*-Value
GFAP	Aβ42/Aβ40	Age	–0.230	0.230
GFAP	Aβ42/Aβ40	Age, Sex	–0.233	0.232
GFAP	pTau181	Age	0.564	0.001
GFAP	pTau181	Age, Sex	0.551	0.002

**Table 3 ijms-26-08134-t003:** Cutoff values, sensitivity, and specificity for Aβ or pTau181 positivity.

	Cut Off	Sensitivity	Specificity
Aβ Positivity	4412.0	63.6% (30.8–89.1%)	89.5% (66.9–98.7%)
pTau Positivity	4412.0	80.0% (44.4–97.5%)	95.0% (75.1–99.9%)

Sensitivity and specificity are presented with 95% CIs shown in parentheses.

## Data Availability

The original contributions presented in the study are included in the article; further inquiries can be directed to the corresponding author.

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
