# Peer review of "Fully Automated Measurement of GFAP in CSF Using the LUMIPULSE® System: Implications for Alzheimer’s Disease Diagnosis and Staging"

_ijms, 2025, doi:10.3390/ijms26178134_

Round 1

Reviewer 1 Report

Comments and Suggestions for Authors

In this article, authors assessed the usefulness of using Lumipulse G GFAP assay with CSF samples as a diagnostic tool for AD. They showed the potential of using this assay alongside other biomarkers such as Aβ40, Aβ42, and pTau181 to differentiate AD from non-AD cases. The findings suggested the utility of this assay in detecting tau-related pathology.

Article needs major revision before accepting it for publication. Discussion didn’t present the comparison of the current study results to those of similar studies in literature. Some subsections of abstract are recommended to be restated to better present the aim and novel insights this study provides. Authors should take into consideration that currently translational research focuses on blood-based biomarkers rather than CSF biomarkers for AD diagnostics because blood collection is more accessible, less invasive, and cost-effectiveness compared to CSF samples. I suggest that authors add a few sentences about that in the introduction.

Abstract

I suggest restating the background and conclusion to better present the aim of study and the importance of its findings.

  • Background didn’t state the aim of assessing this assay in CSF samples. What’s the advantage of Fujirebio Lumipulse G GFAP assay rather than other assays available in market? And why the authors want to assess the assay with CSF samples not just using it with plasma samples. Authors discussed these points in the discussion section. I suggest that authors add one or two sentences in the background to indicate that this study aims to provide insights which may improve AD diagnostic and provide better understanding of disease stage, and how this might affect the treatment follow-up and outcome. Briefly, authors should point in the abstract to how their study’s findings might provide.
  • Lines 29-30: Please restate this sentence “These findings suggest that GFAP may serve as a valuable biomarker for AD”. GFAP has been documented as a valuable biomarker for AD in several studies in literature. I recommend mentioning how GFAP measurements with other biomarkers in this study could provide better understanding of disease pathogenesis, staging, and/or response to treatment.

Introduction

  • As I mentioned earlier blood-based biomarkers for AD diagnostics represent an important area of research currently. I suggest that authors add to the introduction one or two sentences which show the advantage of measuring GFAP in CSF rather than plasma.
  • Line 57: Please delete cerebrospinal fluid and use the abbreviation only. Full name was mentioned earlier in the article.

Materials and methods

  • Lines 87-88: Please use only the abbreviations; MCI and AD. Full names were mentioned earlier in the article.
  • Why didn’t the authors measure pTau 217 in CSF samples? Recently, several studies have shown the superiority of pTau 217 to pTau 181 in CSF and plasma for AD diagnosis. I suggest that authors add the pTau 217 measurements to this study if these measurements are available.

https://pubmed.ncbi.nlm.nih.gov/34310832/

https://alzres.biomedcentral.com/articles/10.1186/s13195-024-01513-9

https://pmc.ncbi.nlm.nih.gov/articles/PMC8711249/

Results

  • Table 1: Can the authors clarify the statistical test used to calculate the p-value provided in the table. Is it ANOVA?
  • Figure 3: What is the cut-off value used to determine the CSF pTau181 positivity?

Discussion

  • Authors discussed their results and thoroughly explained them based on literature, but they did not compare their results to other studies that showed data measuring GFAP, Aβ, and pTau in CSF of AD patients.
  • Lines 206-215: I suggest that authors add a sentence at the end of this paragraph indicating that GFAP is a reliable biomarker for AD diagnosis and staging. Its measurement in plasma or CSF provides information about disease pathology and stage.

Title

Referring to the study findings, I suggest adding “and Staging” at the end of title to be “Fully Automated Measurement of GFAP in CSF Using the LUMIPULSE® System: Implications for Alzheimer’s Disease Diagnosis and Staging”

Comments on the Quality of English Language

Minimal English editing is required.

Author Response

Comments 1: Background didn’t state the aim of assessing this assay in CSF samples. What’s the advantage of Fujirebio Lumipulse G GFAP assay rather than other assays available in market? And why the authors want to assess the assay with CSF samples not just using it with plasma samples. Authors discussed these points in the discussion section. I suggest that authors add one or two sentences in the background to indicate that this study aims to provide insights which may improve AD diagnostic and provide better understanding of disease stage, and how this might affect the treatment follow-up and outcome. Briefly, authors should point in the abstract to how their study’s findings might provide.

Response 1: Thank you for your valuable feedback. In response, we have revised the Background section to clarify the rationale for assessing the Lumipulse G GFAP assay in CSF samples and to highlight the potential clinical significance of our study. Specifically, we have added the following two sentences to lines 12–14 & 15-17:

“Compared to existing assays, the Lumipulse G platform offers high-throughput, rapid, and fully automated quantification of biomarkers, enabling more standardized and accessible clinical study.”

“Assessing GFAP in CSF may provide more direct insights into central nervous system pathology than plasma, and could improve the characterization of Alzheimer’s disease (AD) stages and support treatment monitoring.”

We believe these additions help to clearly state the aim of the study and its implications for AD diagnosis, staging, and potential treatment follow-up.

Comments 2: Lines 29-30: Please restate this sentence “These findings suggest that GFAP may serve as a valuable biomarker for AD”. GFAP has been documented as a valuable biomarker for AD in several studies in literature. I recommend mentioning how GFAP measurements with other biomarkers in this study could provide better understanding of disease pathogenesis, staging, and/or response to treatment.

Response 2: Thank you for your helpful comment. In response, we have revised the sentence to better reflect the current literature and to emphasize the added value of our findings. Specifically, we have added the original sentence with the following to lines 33–36:

“While GFAP has previously been established as a biomarker for AD, our findings highlight that combining GFAP with other biomarkers such as Aβ40, Aβ42, and pTau181 may enhance understanding of AD pathogenesis, disease staging, and possibly treatment response.”

We believe this revision addresses your suggestion and more accurately conveys the implications of our results.

Comments 3: As I mentioned earlier blood-based biomarkers for AD diagnostics represent an important area of research currently. I suggest that authors add to the introduction one or two sentences which show the advantage of measuring GFAP in CSF rather than plasma.

Response 3: Thank you for your insightful comment. In response, we have added the following sentences to the Introduction to highlight the advantages of measuring GFAP in CSF rather than plasma to lines 73–77:

“However, GFAP levels in CSF may provide a more direct representation of central nervous system pathology, especially in the early disease stages, as they are less influenced by peripheral factors affecting blood-based biomarkers. Thus, CSF GFAP may complement plasma measurements by offering greater specificity for brain-derived astrocytic activity.”

We believe this addition helps clarify the rationale for evaluating GFAP in CSF in the context of our study.

Comments 4: Please delete cerebrospinal fluid and use the abbreviation only. Full name was mentioned earlier in the article.

Response 4: Thank you for your comment. As suggested, we have deleted cerebrospinal fluid and use the abbreviation only on line 64.

Comments 5: Lines 87-88: Please use only the abbreviations; MCI and AD. Full names were mentioned earlier in the article.

Response 5: Thank you for your comment. As suggested, we have revised the text on line 247 to use only the abbreviations “CU,” “MCI,” and “AD,” as the full terms were already introduced earlier in the manuscript. Additionally, following the suggestion from the Assistant Editor, we have reorganized the manuscript structure to comply with the journal’s required format: 1. Introduction; 2. Results; 3. Discussion; 4. Materials and Methods; 5. Conclusions. Accordingly, the Materials and Methods section now appears after the Discussion section.

Comments 6: Why didn’t the authors measure pTau 217 in CSF samples? Recently, several studies have shown the superiority of pTau 217 to pTau 181 in CSF and plasma for AD diagnosis. I suggest that authors add the pTau 217 measurements to this study if these measurements are available.

https://pubmed.ncbi.nlm.nih.gov/34310832/

https://alzres.biomedcentral.com/articles/10.1186/s13195-024-01513-9

https://pmc.ncbi.nlm.nih.gov/articles/PMC8711249/

Response 6: Thank you for your comment. Regarding pTau217, we fully acknowledge its emerging importance and superior performance in recent studies for the diagnosis of AD. However, in the present study, we were unable to measure CSF pTau217 due to the limited volume of available CSF samples, which had already been consumed for other biomarker analyses. We agree that the inclusion of pTau217 would strengthen the study and will consider incorporating it in future investigations with newly collected cohorts. To address this point, we have added a corresponding statement in the Limitations section to clarify the absence of pTau217 data and its potential impact on the study.

Comments 7: Table 1: Can the authors clarify the statistical test used to calculate the p-value provided in the table. Is it ANOVA?

Response 7: Thank you for your comment. P-values were calculated using the Kruskal–Wallis test for continuous variables (Age, Aβ40, Aβ42, Aβ42/Aβ40 ratio, pTau181, and GFAP [pg/mL]) and Fisher’s exact test for the categorical variable (sex).

So the following sentence was added at Table 1 on line 100; “P-values: Kruskal–Wallis test for continuous variables; Fisher’s exact test for sex.”

Comments 8: Figure 3: What is the cut-off value used to determine the CSF pTau181 positivity?

Response 8: Thank you for your comments. We used a cut-off value of 56.5 pg/mL for CSF pTau181 positivity, as indicated in the Instructions for Use (IFU) of the Lumipulse G pTau181 kit. We have clarified this cut-off value for CSF pTau181 positivity in the main text on lines 123-125.

Comments 9: Authors discussed their results and thoroughly explained them based on literature, but they did not compare their results to other studies that showed data measuring GFAP, Aβ, and pTau in CSF of AD patients.

Response 9: Thank you for your insightful comments. We have now incorporated a brief comparison of our findings with previous studies reporting CSF GFAP, Aβ, and pTau levels in AD patients, to better contextualize our results on lines 181–187.

Comments 10: Lines 206-215: I suggest that authors add a sentence at the end of this paragraph indicating that GFAP is a reliable biomarker for AD diagnosis and staging. Its measurement in plasma or CSF provides information about disease pathology and stage.

Response 10: Thank you for your comments. We have added a sentence at the end of the paragraph on lines 297–301 to highlight the potential utility of GFAP as a reliable biomarker for AD diagnosis and staging, based on its measurement in both CSF and plasma.

Comments 11: Title

Referring to the study findings, I suggest adding “and Staging” at the end of title to be “Fully Automated Measurement of GFAP in CSF Using the LUMIPULSE® System: Implications for Alzheimer’s Disease Diagnosis and Staging”

Response 11: Thank you for your helpful suggestion regarding the title. As recommended, we have revised the title to:

“Fully Automated Measurement of GFAP in CSF Using the LUMIPULSE® System: Implications for Alzheimer’s Disease Diagnosis and Staging.”

We believe this revised title more accurately reflects the study's objectives and findings.

Reviewer 2 Report

Comments and Suggestions for Authors

Major comments

1. Methods state Wilcoxon “rank-sum” (independent groups), but multiple figure legends specify “Wilcoxon signed-rank test” (paired) with Bonferroni correction—this is inconsistent and likely incorrect for independent groups (CU/MCI/AD; Aβ+/−; pTau+/−). Please standardize and re-run if needed. See p.4, Fig.1 legend lines ~129–133; p.5, Fig.2 legend lines ~137–141; p.5–6, Fig.3 legend lines ~147–153. 

2. The informed ROC confidence interval >1.0 (impossible). Reported AUC 0.86 with 95% CI 0.69–1.03 for pTau positivity exceeds the admissible upper bound (1.0). Please recompute CIs (e.g., DeLong or bootstrapping with stratified resampling) and correct in text/figure. See p.6, lines ~163–176; Fig.5. 

3. You stratify by CSF Aβ42/Aβ40 and pTau181 positivity but do not specify exact cut-points, their origin (manufacturer, literature, site-validated), nor whether they were pre-specified. Provide numeric thresholds, justification, and whether any data-derived cutoffs were used. Report how many positives per group under each criterion. See p.5, lines ~142–149; p.6, lines ~165–175. 

4. Pre-analytical and analytical validation details are incomplete. The authors dilute CSF 10× prior to GFAP analysis; please document: LLOQ/ULOQ in CSF matrix, linearity with dilution, parallelism (spike-recovery), matrix effects, number of freeze–thaw cycles, and whether carryover/hook effect were assessed at high GFAP. Include within-run and between-run precision (CV%) for CSF, not only plasma (if relying on ref [13], summarize CSF-specific performance here). See p.3, lines ~92–105 (assay description). 

5. N=30 is modest for multi-group comparisons and ROC estimation; please provide a brief power/precision rationale, and for ROC, include sensitivity, specificity, Youden’s J, and 95% CIs at the optimal threshold. Consider bootstrap 1000× for AUC CI stability given class imbalance. See p.6–7, ROC and results sections. 

Minor points:

6. Methods—statistics: You mention “Pearson or Spearman” but figures specify Spearman; state a priori rule (Spearman for non-normal distributions) and include exact p-values for correlations in main text. This is in p.3, lines ~106–115; p.6, Fig.4 legend. 

7. Figure aesthetics: Overlaying violin + beeswarm + boxplots is busy for n=30; consider box+points only; ensure “ns/*//*” labels correspond to the corrected test. Figs.1–3. 

8. Table 1: Report exact tests for sex (χ² or Fisher’s exact) and provide medians (Q1–Q3) with units for all biomarkers. Clarify whether p-values are group-wise omnibus (Kruskal–Wallis) with post hoc or pairwise only. p.4, lines ~118–122. 

9. Methods - assay runtime/throughput: You cite ~30 min result time; in Conclusions you claim up to 120 tests/hour—add a reference or calculation (e.g., instrument channels, batch size). p.8, lines ~238–244. 

10. Inconsistencies: “GFAP may serve as a valuable biomarker for AD” (Abstract) should be toned down to “complementary biomarker reflecting astroglial reactivity associated with tau positivity.” p.1, lines ~27–31. 

11. Grammar/typos: “chem-iluminescent” → “chemiluminescent”; “detector antibodies bind to the captured target” (minor style); ensure spacing around units (pg/mL). p.3, lines ~92–105; throughout. 

Author Response

Comments 1: Methods state Wilcoxon “rank-sum” (independent groups), but multiple figure legends specify “Wilcoxon signed-rank test” (paired) with Bonferroni correction—this is inconsistent and likely incorrect for independent groups (CU/MCI/AD; Aβ+/−; pTau+/−). Please standardize and re-run if needed. See p.4, Fig.1 legend lines ~129–133; p.5, Fig.2 legend lines ~137–141; p.5–6, Fig.3 legend lines ~147–153. 

Response 1: We appreciate the reviewer’s careful reading and valuable comments. As correctly pointed out, the Wilcoxon rank-sum test (also known as the Mann–Whitney U test) was used for statistical comparisons between independent groups (CU/MCI/AD; Aβ+/−; pTau+/−), as described in the Methods section. The mention of the Wilcoxon signed-rank test in the figure legends was an error.

We have now corrected the figure legends in Figures 1–3 to accurately reflect the use of the Wilcoxon rank-sum test with Bonferroni correction for multiple comparisons.

Comments 2: The informed ROC confidence interval >1.0 (impossible). Reported AUC 0.86 with 95% CI 0.69–1.03 for pTau positivity exceeds the admissible upper bound (1.0). Please recompute CIs (e.g., DeLong or bootstrapping with stratified resampling) and correct in text/figure. See p.6, lines ~163–176; Fig.5. 

Response 2: Thank you for pointing out this important issue. As suggested, we have recalculated the 95% CIs for the AUC values using bootstrapping with stratified resampling (1,000 x). The previously reported confidence interval that exceeded 1.0 has been corrected.

The revised confidence intervals now fall within the appropriate bounds, and the updated results have been reflected in the abstract (lines 29-31), the main text (lines 169-170), and in Figure 5 (line 174) accordingly.

Comments 3: You stratify by CSF Aβ42/Aβ40 and pTau181 positivity but do not specify exact cut-points, their origin (manufacturer, literature, site-validated), nor whether they were pre-specified. Provide numeric thresholds, justification, and whether any data-derived cutoffs were used. Report how many positives per group under each criterion. See p.5, lines ~142–149; p.6, lines ~165–175. 

Response 3: Thank you for your comment. We have now included the exact cutoff values used to define biomarker positivity in the main text (lines 149–150). The cutoff values used to define biomarker positivity were 0.067 for the CSF Aβ42/Aβ40 ratio, based on a previously published study (Nojima et al., 2022), and 56.5 pg/mL for CSF pTau181, as specified in the manufacturer’s package insert for the Lumipulse G pTau181 assay. These thresholds were pre-specified and not derived from the current dataset. Based on these criteria, 11 out of 30 participants (36.7%) were classified as Aβ-positive, and 10 participants (33.3%) as pTau181-positive (lines 123–127).

Comments 4: Pre-analytical and analytical validation details are incomplete. The authors dilute CSF 10× prior to GFAP analysis; please document: LLOQ/ULOQ in CSF matrix, linearity with dilution, parallelism (spike-recovery), matrix effects, number of freeze–thaw cycles, and whether carryover/hook effect were assessed at high GFAP. Include within-run and between-run precision (CV%) for CSF, not only plasma (if relying on ref [13], summarize CSF-specific performance here). See p.3, lines ~92–105 (assay description). 

Response 4: Thank you for your thorough and constructive comments. We have added detailed information on the pre-analytical and analytical aspects of the GFAP assay in CSF samples to the Materials and Methods section (lines 264–273). Additionally, following the suggestion from the Assistant Editor, we have reorganized the manuscript structure to comply with the journal’s required format: 1. Introduction; 2. Results; 3. Discussion; 4. Materials and Methods; 5. Conclusions. Accordingly, the Materials and Methods section now appears after the Discussion section.

Comments 5: N=30 is modest for multi-group comparisons and ROC estimation; please provide a brief power/precision rationale, and for ROC, include sensitivity, specificity, Youden’s J, and 95% CIs at the optimal threshold. Consider bootstrap 1000× for AUC CI stability given class imbalance. See p.6–7, ROC and results sections. 

Response 5: Thank you for your valuable comment. We agree that the sample size of N = 30 is modest for multi-group comparisons and ROC analysis. Our study was designed as an exploratory investigation to identify preliminary trends and generate hypotheses for future research.

To address your concern, we conducted post hoc power analyses using G*Power (version 3.1.9.7). Assuming a two-tailed t-test, an alpha level of 0.05, and a medium-to-large effect size (Cohen’s d = 0.65), the achieved statistical power for each key comparison was as follows:

Aβ-positive (n = 11) vs. Aβ-negative (n = 19): power ≈ 0.381 (38.1%)

pTau181-positive (n = 10) vs. pTau181-negative (n = 20): power ≈ 0.367 (36.7%)

These results confirm that the statistical power is limited, which we acknowledge as a limitation of the present study. Nevertheless, we believe that the findings offer meaningful preliminary insights and can serve as a basis for hypothesis generation and the design of adequately powered future studies. Accordingly, we have added a brief rationale regarding statistical precision and sample size limitations in the revised Materials and Methods (lines 283–286) and Limitations sections (lines 212–220).

In response to your suggestion, we conducted ROC analyses with 1,000× bootstrap replicates to ensure the stability of the AUC confidence intervals, particularly considering the class imbalance (Figure 5). Additionally, we have added a new table (Table 2) summarizing the optimal cutoff value determined by Youden’s index, along with sensitivity, specificity, and their respective 95% confidence intervals at the optimal threshold. These details have been incorporated into the revised Results section (lines 151–153) and the newly added Table 2 (lines 161–162).

We appreciate your guidance in improving the robustness and transparency of our analysis.

Minor points:

Comments 6: Methods—statistics: You mention “Pearson or Spearman” but figures specify Spearman; state a priori rule (Spearman for non-normal distributions) and include exact p-values for correlations in main text. This is in p.3, lines ~106–115; p.6, Fig.4 legend. 

Response 6: Thank you for this important comment. We have revised the Materials and Methods section to remove the reference to Pearson correlation. We have clarified that Spearman’s rank correlation was used in this study due to the non-normal distribution of the biomarker data. Furthermore, as requested, we have now included the exact p-values for the correlation analyses directly in the main text (lines 137-138).

Comments 7: Figure aesthetics: Overlaying violin + beeswarm + boxplots is busy for n=30; consider box+points only; ensure “ns/*//*” labels correspond to the corrected test. Figs.1–3. 

Response 7: Thank you for your helpful suggestion regarding the figure design and statistical annotations. We have removed the violin plots from Figures to improve clarity given the sample size (n = 30). The updated figures now include only boxplots with individual data points. Additionally, we have re-performed the statistical significance testing to ensure accuracy, and all “ns/*//*” labels now reflect the corrected p-values based on the updated analyses.

Comments 8: Table 1: Report exact tests for sex (χ² or Fisher’s exact) and provide medians (Q1–Q3) with units for all biomarkers. Clarify whether p-values are group-wise omnibus (Kruskal–Wallis) with post hoc or pairwise only. p.4, lines ~118–122. 

Response 8: Thank you for your valuable comment. We have revised Table 1 to include the exact statistical test used for sex comparison (Fisher’s exact test) and the p-values for continuous variables were calculated using the Kruskal–Wallis test as a group-wise omnibus test.

We have also updated the presentation of all biomarkers to report the median and interquartile range (Q1–Q3), along with corresponding units for each value (pg/mL).

The revised text and table have been updated accordingly.

Comments 9: Methods - assay runtime/throughput: You cite ~30 min result time; in Conclusions you claim up to 120 tests/hour—add a reference or calculation (e.g., instrument channels, batch size). p.8, lines ~238–244.

Response 9: Thank you for this important point. We have now added an appropriate reference to support our statement regarding the assay throughput, citing M. R. de Rancher et al. (2016) on line 291. We hope this clarification adequately addresses the reviewer’s concern. Thank you again for your valuable feedback.

Comments 10: Inconsistencies: “GFAP may serve as a valuable biomarker for AD” (Abstract) should be toned down to “complementary biomarker reflecting astroglial reactivity associated with tau positivity.” p.1, lines ~27–31. 

Response 10: Thank you for this helpful suggestion. As recommended, we have revised the sentence in the Abstract to tone down the description of GFAP. The phrase (lines 36-37):

“GFAP may serve as a valuable biomarker for AD”

has been changed to:

“GFAP may serve as a complementary biomarker reflecting astroglial reactivity associated with tau positivity,”

to more accurately reflect its role in the context of AD pathology.

Comments 11: Grammar/typos: “chem-iluminescent” → “chemiluminescent”; “detector antibodies bind to the captured target” (minor style); ensure spacing around units (pg/mL). p.3, lines ~92–105; throughout. 

Response 11: Thank you for pointing out these editorial issues. Regarding the appearance of occasional irregular hyphens (e.g., “chem-iluminescent”), we would like to clarify that these are due to the journal’s manuscript formatting system and were not intentionally inserted.

We have carefully reviewed the manuscript and have corrected the spacing around units (e.g., “pg/mL”) by adding appropriate spaces between numeric values and their corresponding units, in accordance with standard scientific writing conventions.

Round 2

Reviewer 1 Report

Comments and Suggestions for Authors

Thank you for addressing my comments of the first revision.

Author Response

Comments 1: Thank you for addressing my comments of the first revision.

Response 1: We thank the Reviewer 1 for their positive feedback and are glad that the previous revisions have addressed your concerns.

Reviewer 2 Report

Comments and Suggestions for Authors

Thank you for the revisions. Some aspects are still pending. Please:

(i) add partial correlations adjusting at least for age (±sex) and clarify the A/T/N framing (N not assessed).

After this aclaration/modifications the manuscript is suitable.

Author Response

Comments 1: Thank you for the revisions. Some aspects are still pending. Please:

(1) add partial correlations adjusting at least for age (±sex) and (2) clarify the A/T/N framing (N not assessed).

After this aclaration/modifications the manuscript is suitable.

Response 1: Thank you for your comment.

For (1): As suggested, we conducted partial correlation analyses adjusting for age and sex. As described in the revised manuscript, partial correlation analyses demonstrated no significant association between GFAP and the Aβ42/Aβ40 ratio after adjusting for age alone (r = –0.230, p = 0.230) or for both age and sex (r = –0.233, p = 0.232). In contrast, GFAP was significantly positively correlated with pTau181 after adjusting for age (r = 0.564, p = 0.001) and also when adjusting for both age and sex (r = 0.551, p = 0.002).

These results have been added to Table 2 and are described in the Results section (lines 144–150) and the Discussion section (lines 197–199). Corresponding revisions have also been made to the Methods section (lines 292–294).

For (2): Regarding the A/T/N classification, we have clarified in the revised manuscript that the “N” (neurodegeneration) marker was not assessed in this study due to limited CSF sample volume. This limitation has been explicitly acknowledged in the Limitations section (lines 246–252).

We appreciate your thoughtful review and hope that the revised manuscript meets your expectations.